# Incidence, risk factors, and prognosis of acute kidney injury in hospitalized patients with acute cholangitis

**Tae Won Lee[1], Wooram Bae[1], Seongmin Kim[1], Jungyoon Choi[1], Eunjin Bae[1,2,3], Ha Nee Jang[4], Se-Ho Chang[2,3,4], Dong Jun Park**[1,2,3] *

1 Department of Internal Medicine, Gyeongsang National University Changwon Hospital, Changwon, South Korea, 2 Department of Internal Medicine, Gyeongsang National University College of Medicine, Jinju, South Korea, 3 Institute of Health Science, Gyeongsang National University, Jinju, South Korea, 4 Department of Internal Medicine, Gyeongsang National University Hospital, Jinju, South Korea

* drpdj@naver.com

**Data Availability Statement:** All relevant data are within the paper and its Supporting information files.

## Abstract

### Background

The association between acute cholangitis (AC) and acute kidney injury (AKI) remains unclear. We investigated the incidence, and clinical course of AKI in patients with AC, and the long-term prognosis.

### Methods

We performed a single-center retrospective study of patients hospitalized with AC in a tertiary care center from January 2011 to December 2017. The risk factors for AKI were evaluated, and AKI severity was analyzed using the Systemic Inflammatory Response System (SIRS), quick sequential organ failure assessment (qSOFA) score, and 2018 Tokyo Guidelines (TG) grade. To calculate the relative risk of death based on AKI, hazard ratios (HRs) and 95% confidence intervals (CIs) were obtained using Cox's proportional hazard models.

### Results

A total of 1,438 patients with AC were included, of whom 18.2% (n = 261) developed AKI. AKI patients were older, and had a lower systolic blood pressure and more comorbidities including hypertension (HT), chronic kidney disease, and cardiovascular accidents. Disease severity (as assessed by SIRS, qSOFA, and the Tokyo Guidelines grade) was higher in the AKI group, as was the in-hospital mortality rate. Multivariate analysis revealed that age, HT, SIRS and qSOFA scores $\geq 2$, and TG grade of III were significant risk factors for AKI. Kaplan-Meier analysis revealed significantly higher mortality in the AKI than non-AKI group. AKI (HR = 1.853; 95% CI: 1.115–3.079) and TG grade III (HR = 2.139; 95% CI: 1.190–3,846) were independent predictors of all-cause AC mortality, even after adjusting for all covariates. The annual rate of decline in the estimated glomerular filtration rate was faster in the AKI than non-AKI group ($2.9 \pm 6.7$ vs. $0.5 \pm 5.3$ mL/min/1.73 m$^2$/year, $p < 0.001$).

**Funding:** The authors received no specific funding for this work.

**Competing interests:** The authors have declared that no competing interests exist.

## Conclusions

AKI development increased AC severity and mortality. Our results suggest that clinicians should monitor AKI status and perform appropriate management as soon as possible.

## 1. Introduction

Acute cholangitis (AC) is usually defined as a bacterial infection of the biliary tract caused by a lithiasic or tumoral obstruction [1, 2]. Intensive care unit (ICU) admission of AC patients is relatively common; AC accounts for 5% of all septic shock cases [3] and 10–29% of intraabdominal sepsis cases [4–6]. The mortality rate is 5–10% despite resuscitation, early administration of broad-spectrum antibiotics, and prompt decompression [7]. Early diagnosis and recognition of severe presentations is essential to allow clinicians to initiate appropriate management [8]. To this end, the recommendations of the Tokyo Guidelines (TG) Working Group have been regularly updated from 2007 to 2018, i.e., since an International Consensus Meeting in 2006 focusing on the diagnosis, severity assessment, and management of AC [9–13].

Acute kidney injury (AKI) is significantly associated with mortality, the length of hospital stay, and healthcare costs [14]. AKI is an important issue in patients with various infectious diseases. The incidence of AKI ranges from 5% to 51% in patients with sepsis [15–19]. AKI is an independent poor prognostic factor for sepsis. Many studies have shown that several factors contribute to AKI development and outcomes in septic patients [20–22]. In 2012, the Kidney Disease: Improving Global Outcomes (KDIGO) consortium developed AKI clinical practice guidelines based on a careful review of the evidence [23]. The guidelines include chapters on AKI definition, staging, risk assessment, evaluation, prevention, and treatment. AKI according to the guidelines is defined as any of the following: Increase in serum creatinine (SCr) by X0.3 mg/dl within 48 hours; or Increase in SCr to X1.5 times baseline, which is known or presumed to have occurred within the prior 7 days; or Urine volume 0.5 ml/kg/h for 6 hours. Although two very old studies on renal failure associated with AC have appeared [24, 25], no study has applied the KDIGO guidelines to evaluate the incidence, risk factors, or clinical outcomes of AKI in association with AC, which was the goal of this study.

## 2. Materials and methods

### Study population

We retrospectively studied 1,438 AC patients aged $\geq$ 18 years admitted to Gyeongsang National University Hospital between January 2011 and December 2017. Follow-up ended on December 31, 2020. AC patients were identified by searching the hospital database for ICD-10-CM codes K803 and K830. Information on demographic and clinical characteristics, laboratory findings, and comorbidities obtained at the time of admission, and in the outpatient clinic after discharge, were retrieved from electronic medical records (EMRs). The exclusion criteria were as follows: incomplete medical records, end-stage renal disease (ESRD), and/or emergency department treatment only.

### Definition and clinical assessments

According to KDIGO (23), AKI is defined when either of the following is present: an increase in serum creatinine > 0.3 mg/dL within 48 h, or a serum creatinine level > 1.5-fold that of the

baseline that is known or presumed to have occurred within the last 7 days. Criteria involving the urine volume were excluded because the volume was not measured in the large majority of patients admitted to non-ICU wards. The baseline creatinine level was that recorded within 3 months before admission. In the absence of such data, the baseline creatinine level was estimated by consecutive blood tests during hospitalization (as described in the KDIGO guidelines). The AKI stage was evaluated according to the KDIGO guideline (23). The estimated glomerular filtration rate (eGFR) was calculated using the Modification of Diet in Renal Disease formula [$1.86 \times$ (plasma creatinine level)$- 1.154 \times$ (age)$- 0.203$)] $\times$ (0.74 if female) $\times$ (1.210 if black). Creatinine was measured using the Jaffe method. Chronic kidney disease (CKD) was defined as an eGFR $< 60$ mL/min/1.73 m$^2$. Three decompression methods were used: surgical, endoscopic, and percutaneous transhepatic biliary drainage (PTBD). Endoscopic decompression involved placement of a naso-biliary tube and a biliary stent. AC severity was assessed using the Systemic Inflammatory Response System (SIRS) [26], quick Sequential Organ Failure Assessment (qSOFA) [27, 28], and TG grade [12]. The primary outcome was the incidence of AKI in AC patients. We sought to identify clinical and demographic variables predictive of AKI in AC patients. We explored the associations of SIRS and qSOFA scores $\geq 2$ and the Tokyo Guidelines grade with AKI severity (in line with the KDIGO guidelines), and how AKI affected in-hospital mortality and 1-year mortality of AC patients. Finally, in patients whose renal function was assessed $> 1$ year after discharge, the annual decline in function was compared between the AKI and non-AKI groups. The Institutional Review Board (IRB) of Gyeongsang National University Hospital (GNUH) approved this study (IRB no. 2020-05-008) and waived the need for informed consent due to its retrospective study design.

## Statistical analysis

Data are presented as means ± standard deviations or as frequencies (counts and percentages). The normality of continuous variables was examined using the Shapiro-Wilk test. When a normal distribution was confirmed, the independent t-test was used to analyze the data, which are shown as mean ± standard deviation. Continuous variables with non-normal distributions were compared using the Mann-Whitney U test. Categorical variables were compared by the chi-squared or Fisher exact test. We used multivariate logistic regression used to identify significant risk factors for AKI. Variables that were significant in univariate logistic analysis were included in multivariate analysis, and the variables of the final predictive model were selected by the backward elimination method. Internal validity of the final multivariable model was assessed using repeated k-fold cross validation with 10 fold, 10 repeated, 75% train dataset and 25% test dataset of the total dataset. Next, logistic regression coefficients were employed to develop an equation for predicting AKI. Finally, receiver operating characteristic (ROC) curves were constructed to determine a threshold with optimal sensitivity and specificity. Also calibration of the final multivariable model was assessed based on the Hosmer–Lemeshow goodness-of-fit test and, visually, by plotting predicted probability of AKI development against the observed rate of the AKI in each decile. To explore the association between AKI and all-cause mortality, Kaplan-Meier curves were plotted by AKI status. To calculate the relative risks of death, hazard ratios (HRs) with 95% confidence intervals (CIs) were derived based on Cox's proportional hazards models. All variables with $p$-values $< 0.05$ were included in the multivariate analysis. All statistical analyses were performed with SPSS for Windows software (ver. 24.0; SPSS Inc., Chicago, IL, USA) and R software (version 4.02; R Core Team,2020) (R: Language and Environment for Statistical Computing. R Foundation for Statistical Computing, Vienna, Austria).

## 3. Results

### Clinical and laboratory characteristics by AKI status

From January 1 2011 to December 31 2017, 1,438 patients with AC were enrolled in this study; 488 were excluded. We divided all patients into AKI and non-AKI groups (Fig 1). Table 1 lists the baseline characteristics of all patients diagnosed with AC. The mean age was 70.9 years and 57% (n = 820) were male. AKI developed in 261 of the 1,438 patients. AKI patients were significantly older than non-AKI patients (74.8 ± 11.3 vs. 70.0 ± 13.4 years, $p < 0.001$). AKI patients exhibited higher rates of hypertension (HT) and CKD (48.3 vs. 32.4%, $p < 0.001$ and 11.9 vs. 7.1%, $p = 0.015$, respectively). The AC etiologies did not differ between the groups. Surgical decompression was more common in the non-AKI group (15.7 vs. 9.2%, $p = 0.007$) and PTBD was more preferred in the AKI group (14.8 vs. 24.9%, $p < 0.001$). The disease severity index (derived using the Tokyo Guidelines grade, and SIRS and qSOFA scores) correlated positively with AKI development ($p < 0.001$). The total white blood cell, bilirubin, and C-reactive protein levels were significantly higher in the AKI group (all $p < 0.001$). The bile culture-positive rate was higher in the AKI group ($p = 0.027$) but the blood-culture rate did not differ significantly between the two groups. Patients with AC were hospitalized for a mean of 10.6 ± 9.3 days. Moreover, patients in the AKI group had a significantly longer hospital stay than those in the non-AKI group (13.3 ± 11.3 vs. 10.0 ± 8.7 days, $p < 0.001$). The in-hospital mortality rate was much higher for the AKI group [13 (5.0%) vs. 12 (1.0%), $p < 0.001$].

### The incidence and independent risk factors of AKI in AC patients

As defined by the KDIGO guidelines, 261 (18.2%) patients developed AKI: 146 (55.9%) stage 1, 74 (28.4%) stage 2, and 41 (15.7%) stage 3 AKI. Table 2 shows the AKI-associated clinical variables. Age, HT, CKD, a previous cardiovascular accident (CVA), Tokyo Guidelines grade III, SIRS and qSOFA scores ≥ 2, and a positive bile culture were associated with AKI. Multivariate analysis revealed that age (HR = 1.021; 95% CI: 1.007–1.036, $p = 0.004$), HT (HR = 1.762; 95% CI: 1.277–2.430, $p = 0.001$), TG grade III (HR = 10.839; 95% CI: 7.414–15.846, $p < 0.001$), a SIRS score ≥ 2 (HR = 2.334; 95% CI: 1.575–3.456, $p < 0.001$), and a qSOFA score ≥ 2 (HR = 1.872; 95% CI: 1.271–2.757, $p < 0.001$) were independent risk factors for AKI development in AC patients (Table 2).

### Relationships of the TG grade, SIRS, and qSOFA scores with AKI status

Severe AKI was significantly more common in AC patients with a higher Tokyo Guidelines grade ($p < 0.001$). SIRS and qSOFA scores ≥ 2 were strongly associated with more severe AKI ($p = 0.006$ and $p = 0.002$, respectively) (Table 3).

### AKI prediction in AC patients

The regression coefficients (standard error) for age, HT, TG grade III, SIRS score ≥ 2, and qSOFA score ≥ 2 for the multiple logistic regression model used to predict the risk of AKI were 0.016 (0.007), 0.592 (0.163), 2.420 (0.201), 0.914 (0.199), and 0.658 (0.196), respectively. Therefore, the regression equation for AKI prediction was as follows: AKI prediction score = (0.02 × age) + (0.6 × HT) + (2.4 × TG grade III) + (0.9 × SIRS score) + (0.7 × qSOFA score ≥ 2). This equation yields a numerical score predicting AKI in AC patients. The cutoff (determined using ROC curves) was ≥ 1.9 (area under the curve = 0.798; 95% CI: 0.767–0.828; sensitivity = 70.1%, specificity = 77.4%) (Fig 2).

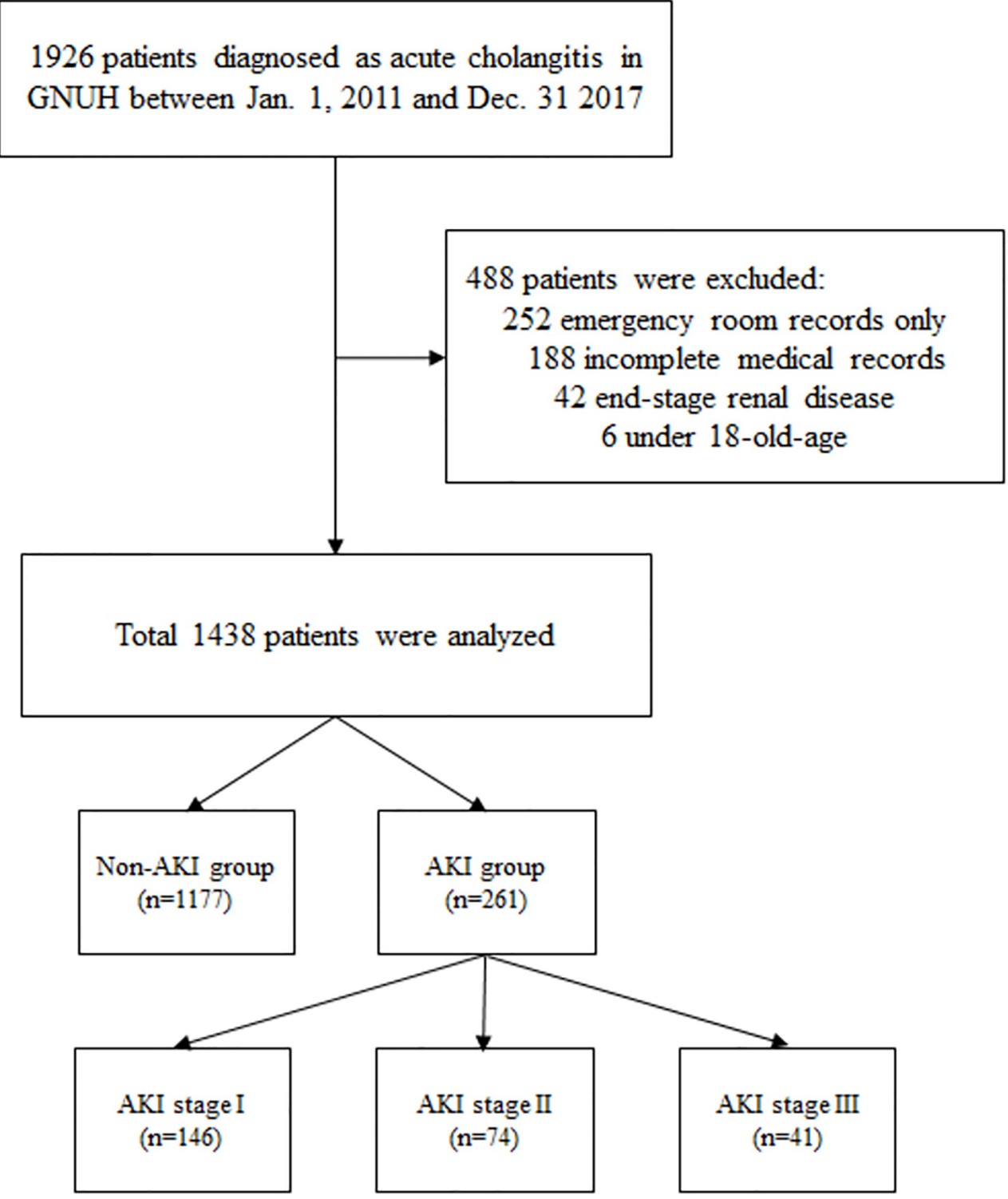

**Fig 1. Overall workflow of patient's enrollment.**

**Table 1. Baseline characteristics for acute kidney injury in patients with acute cholangitis.**

| | Total (N = 1438) | Non-AKI (N = 1177) | AKI (N = 261) | *P* value |
|---|---|---|---|---|
| Age (yr) | 70.9 ± 13.1 | 70.0 ± 13.4 | 74.8 ± 11.3 | < 0.001 |
| Female, n (%) | 618 (43.0) | 519 (44.1) | 99 (37.9) | 0.069 |
| SBP, mmHg | 121.0 ± 18.2 | 122.3 ± 17.8 | 115.1 ± 18.9 | < 0.001 |
| Body temperature, ˚C | 36.6 ± 0.5 | 36.5 ± 0.5 | 36.7 ± 0.7 | 0.004 |
| Comorbidities | | | | |
| Hypertension, n (%) | 507 (35.3) | 381 (32.4) | 126 (48.3) | < 0.001 |
| Diabetes, n (%) | 292 (20.3) | 229 (19.5) | 63 (24.1) | 0.089 |
| CKD, n (%) | 84 (5.8) | 59 (5.0) | 25 (9.6) | 0.008 |
| Liver cirrhosis, n (%) | 68 (4.7) | 54 (4.6) | 14 (5.4) | 0.593 |
| Heart failure, n (%) | 30 (2.1) | 21 (1.8) | 9 (3.4) | 0.089 |
| OMI, n (%) | 27 (1.9) | 19 (1.6) | 8 (3.1) | 0.118 |
| CVA, n (%) | 59 (4.1) | 42 (3.6) | 17 (6.5) | 0.030 |
| Etiology, n (%) | | | | |
| Stone | 438 (30.5) | 364 (30.9) | 74 (28.4) | 0.414 |
| Tumor | 196 (13.6) | 159 (13.5) | 37 (14.2) | 0.776 |
| Decompression, n (%) | | | | |
| Endoscopic | 410 (28.5) | 323 (27.4) | 87 (33.3) | 0.076 |
| Surgical | 209 (14.5) | 185 (15.7) | 24 (9.2) | 0.007 |
| Percutaneous | 239 (16.6) | 174 (14.8) | 65 (24.9) | < 0.001 |
| Severity assessment | | | | |
| TOKYO grade | | | | < 0.001 |
| I | 927 (64.5%) | 852 (72.4%) | 75 (28.7%) | |
| II | 326 (22.7%) | 246 (20.9%) | 80 (30.7%) | |
| III | 185 (12.9%) | 79 (6.7%) | 106 (40.6%) | |
| SIRS ≥ 2, n (%) | 247 (17.2) | 146 (12.4) | 101 (38.7) | < 0.001 |
| qSOFA score ≥ 2, n (%) | 262 (18.2) | 159 (13.5) | 103 (39.5) | < 0.001 |
| Laboratory findings | | | | |
| WBC, $10^9$/L | 10.7 ± 5.9 | 9.9 ± 5.0 | 14.0 ± 7.8 | < 0.001 |
| Hemoglobin, g/dL | 12.4 ± 1.9 | 12.5 ± 1.8 | 12.0 ± 1.9 | < 0.001 |
| Platelet, $10^9$/L | 210.0 ± 95.0 | 219.9 ± 92.7 | 165.6 ± 92.6 | < 0.001 |
| Creatinine, mg/dL | 0.96 ± 0.57 | 0.80 ± 0.27 | 1.69 ± 0.93 | < 0.001 |
| Total bilirubin, mg/dL | 3.2 ± 3.5 | 2.9 ± 3.3 | 4.3 ± 4.2 | < 0.001 |
| AST, U/L | 172.9 ± 258.1 | 163.8 ± 220.5 | 213.7 ± 382.3 | 0.043 |
| ALT, U/L | 138.9 ± 161.4 | 138.8 ± 157.8 | 139.5 ± 176.8 | 0.943 |
| Albumin, g/dL | 3.7 ± 0.6 | 3.7 ± 0.6 | 3.3 ± 0.6 | < 0.001 |
| CRP, mg/L | 69.3 ± 76.3 | 55.0 ± 63.7 | 132.4 ±93.6 | < 0.001 |
| Positive culture | | | | |
| Blood | 204 (14.2) | 167 (14.2) | 37 (14.2) | 0.996 |
| Bile | 343 (23.9) | 267 (22.7) | 76 (29.1) | 0.027 |
| Hospital stay (days) | 10.6 ± 9.3 | 10.0 ± 8.7 | 13.3 ± 11.3 | < 0.001 |
| In-hospital mortality, n (%) | 25 (1.7) | 12 (1) | 13 (5) | < 0.001 |

SBP, systolic blood pressure; CKD, chronic kidney disease; OMI, old myocardial infarction; CVA, cerebral vascular accident; WBC, white blood cell; AST, aspartate transaminase; ALT, alanine transaminase; CRP, c-reactive protein.

## Validation of AKI prediction model

Repeated K-fold cross validation was performed for internal validation of AKI prediction model. We divided our patients into two datasets (75% train dataset versus 25% test dataset).

**Table 2. Independent risk factors for the development of AKI.**

| | Univariate logistic regression | | Multivariate logistic regression | |
|---|---|---|---|---|
| | OR (95%CI) | *P* value | OR (95%CI) | *P* value |
| Age | 1.032 (1.020–1.044) | < 0.001 | 1.021 (1.007–1.036) | 0.004 |
| Hypertension | 1.950 (1.486–2.559) | < 0.001 | 1.762 (1.277–2.430) | 0.001 |
| CVA | 1.883 (1.054–3.363) | 0.033 | 0.944 (0.472–1.886) | 0.870 |
| Underlying CKD | 1.777 (1.148–2.748) | 0.010 | 0.635 (0.370–1.089) | 0.099 |
| TOKYO Grade III | 14.518 (10.194–20.677) | < 0.001 | 10.839 (7.414–15.846) | < 0.001 |
| SIRS | 4.458 (3.291–6.038) | < 0.001 | 2.334 (1.575–3.456) | <0.001 |
| qSOFA $\geq$ 2 | 3.943 (2.929–5.308) | < 0.001 | 1.872 (1.271–2.757) | 0.002 |
| Positive bile culture | 1.400 (1.037–1.890) | 0.028 | 1.344 (0.945–1.912) | 0.100 |

CVA, cerebral vascular accident; CKD, chronic kidney disease.

As a result, the model accuracy in train and test is 0.846, which is considered a relatively valid model. The risk prediction model showed good calibration, with reasonable agreement between observed and predicted AKI outcome in our AC patients (S1 Fig).

## Prediction of all-cause mortality from AKI

The early hospital mortality rate was higher in the AKI than non-AKI group (1% vs. 5%, $p < 0.001$) (Table 1). During a mean follow-up of 24.5 months, 176 (12.2%) patients died. The association of AKI with all-cause mortality was evaluated using Kaplan-Meier analysis (Fig 3). All-cause mortality in the AKI group was higher than in the non-AKI group ($p < 0.001$). We then performed Cox's regression analyses. In multivariate analysis, AKI (HR = 1.860; 95% CI: 1.338–2.586) was an independent predictor of all-cause mortality in AC patients, even after adjusting for all covariates (Table 4). In addition, AC TG grade III (HR = 2.302; 95% CI: 1.338–2.586) was an independent predictor of all-cause mortality, even after adjusting for several covariates (Table 4).

## Effects of AKI on long-term renal function

To evaluate eGFR decline in the two groups, 878 patients whose GFRs were estimated in the outpatient clinic for at least 1 year were analyzed. During a mean follow-up of 3.4 ± 2.3 years, the annual rate of decline was faster in the AKI than non-AKI group (2.9 ± 6.7 vs. 0.5 ± 5.3 mL/min/1.73 m$^2$/year, $p < 0.001$).

**Table 3. The relationship between TOKYO grade, SIRS, and qSOFA and AKI stage determined by KDIGO guideline.**

| | Stage I (N = 146) | Stage II (N = 74) | Stage III (N = 41) | *P* value |
|---|---|---|---|---|
| **TOKYO grade** | | | | < 0.001 |
| Grade I, n (%) | 59 (78.7%) | 15 (20.0%) | 1 (1.3%) | |
| Grade II, n (%) | 56 (70.0%) | 19 (23.8%) | 5 (6.3%) | |
| Grade III, n (%) | 31 (29.2%) | 40 (37.7%) | 35 (33.0%) | |
| **SIRS** | | | | 0.006 |
| < 2, n (%) | 102 (63.8%) | 38 (23.8%) | 20 (12.5%) | |
| $\geq$ 2, n (%) | 44 (43.6%) | 36 (35.6%) | 21 (20.8%) | |
| **qSOFA** | | | | 0.002 |
| < 2, n (%) | 102 (64.6%) | 35 (22.2%) | 21 (13.3%) | |
| $\geq$ 2, n (%) | 44 (42.7%) | 39 (37.9%) | 20 (19.4%) | |

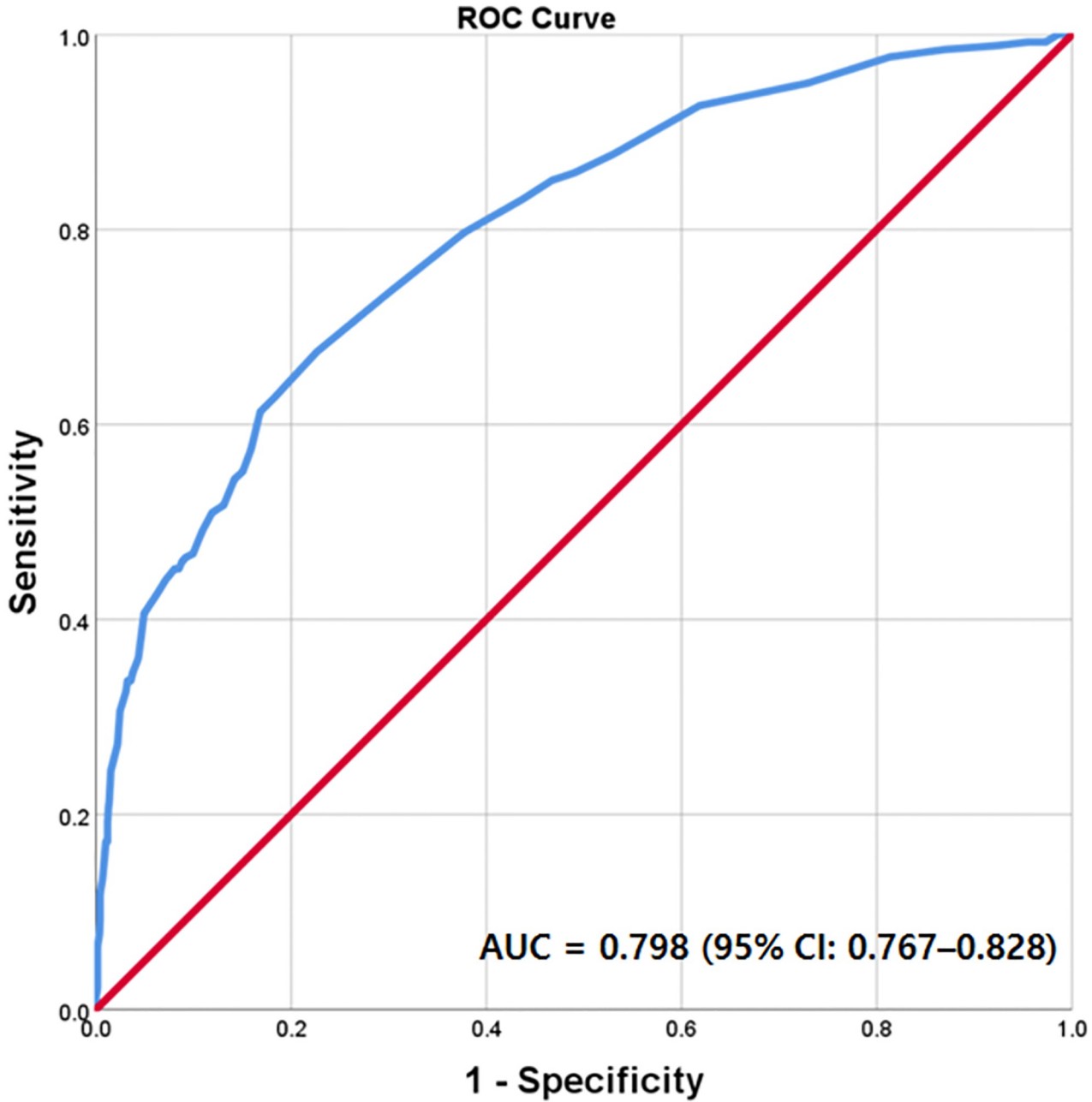

**Fig 2. Receiver operating characteristic (ROC) curves of AKI prediction score to predict AKI in patient with acute cholangitis.**

## 4. Discussion

In this retrospective study, the AKI incidence rate was 18.2%. In the KDIGO guidelines the proportion of AKI in stage I, II, and III disease are 55.9%, 28.4%, and 15.7% respectively. Independent risk factors for AKI associated with AC in this study (as revealed by multivariate analysis) were older age, HT, Tokyo Guidelines grade III, and SIRS and qSOFA scores ≥ 2. AKI severity was positively correlated was AC severity, as assessed by the TG grade, SIRS, and qSOFA. During follow-up, the all-cause mortality rate was higher in the AKI than non-AKI

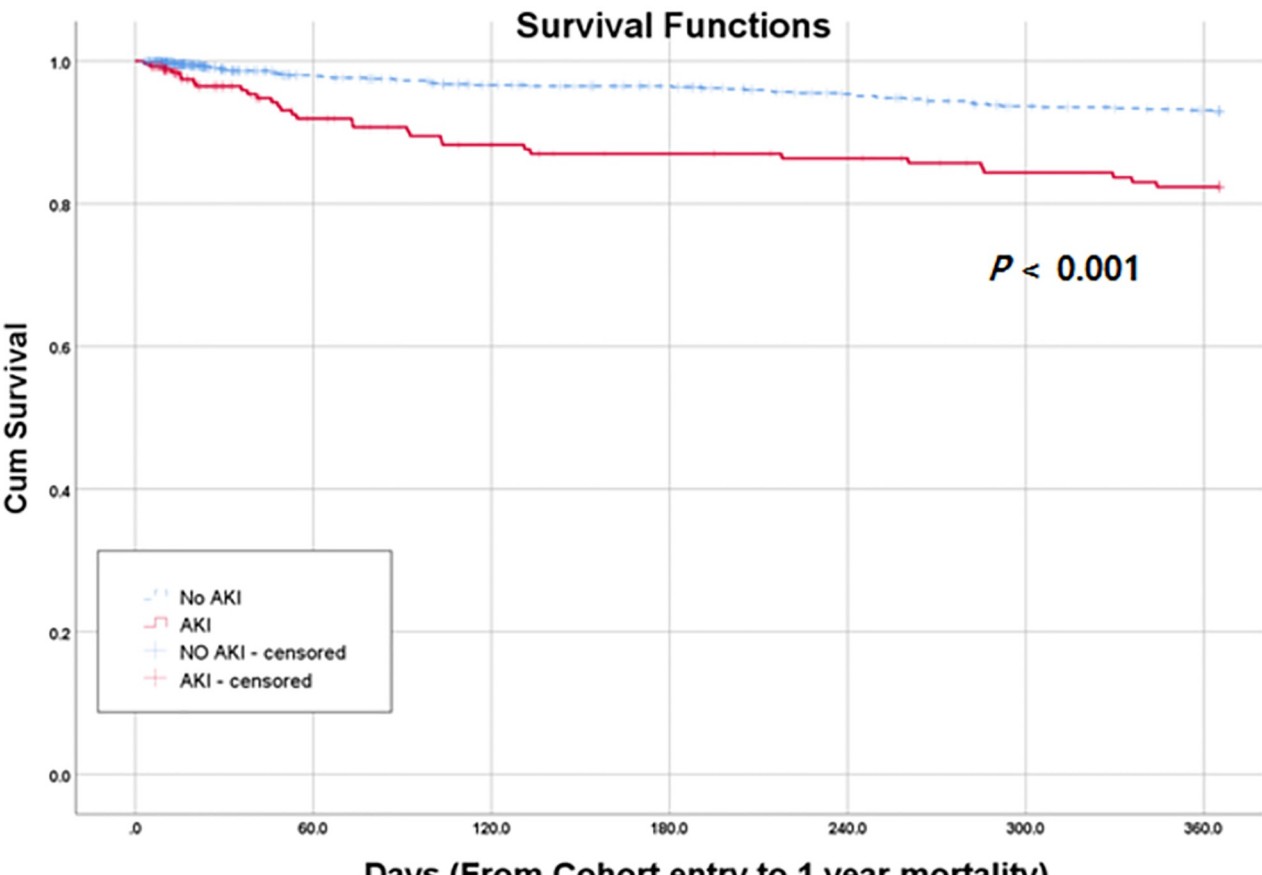

**Fig 3. Kaplan survival analysis of survival associated with the presence of AKI in HD patients.**

group, and AKI was an independent predictor of all-cause mortality in AC patients, even after adjusting for all covariates. In addition, the eGFR decline was faster in the AKI group.

The association between obstructive jaundice and acute renal failure (ARF) is well-recognized; ARF may develop in patients with jaundice caused either by operation or septicemia (or both), but is rarely described [25]. Bismuth et al. (1975) reported an association between AC

**Table 4. Hazard ratios for all-cause mortality risk factors in Acute cholangitis patients.**

|  | All-cause mortality | |
|---|---|---|
|  | **HR (95% CI)** | ***P*** |
| TOKYO classification |  |  |
| Grade I | 1.00 (ref) |  |
| Grade II | 1.222 (0.708–2.111) | 0.471 |
| Grade III | 2.139 (1.190–3.846) | 0.011 |
| Acute kidney injury |  |  |
| No AKI | 1.00 (ref) |  |
| AKI | 1.853 (1.115–3.079) | 0.017 |

HR; hazard ratio, CI; confidence interval. Adjusted for age, SBP, hypertension, CVA, etiology, positive bile culture, decompression treatment, underlying CKD.

and ARF in a large-scale trial [24], which included 283 hospitalized patients with confirmed cholangitis. Twenty-one patients (7.4%) rapidly developed renal insufficiency. ARF was defined as a blood urea nitrogen level > 40 mg/dL or a serum creatinine level > 2.0 mg/dL with a urine:blood urea ratio < 8. Six patients showed mild renal failure, while in two it was severe, and in the remainder moderate. In the present study, the AKI rates in stage I, II, and III disease were 56%, 28%, and 16%, respectively. The higher rate of AKI in our cohort may reflect more strict application of diagnostic criteria for early detection and treatment.

Sepsis is influenced by pathogenic and host factors that evolve over time, and is the primary cause of death from infection [30]. Sepsis was an important risk factor for AKI in several studies [21, 30, 31]. The SIRS criteria have been used since 1991 to classify sepsis [29], but lack the specificity required to diagnose systemic inflammation caused by infection. However, they have high sensitivity [32]. A SIRS score ≥ 2 was a significant independent risk factor for AKI in the present study. The qSOFA can quickly identify infected patients but requires clinical sufficient data [30]. Although the qSOFA score is not part of the new sepsis definition, it is an important part of sepsis work-up in the emergency room [31, 32]. A qSOFA score ≥ 2 was also an independent risk factor for AKI in this study, and correlated significantly with the KDIGO AKI stage.

The diagnostic and severity grades of the 2018 TG (TG18) are preferred worldwide for AC management. According to TG18, the 30-day mortality rate of patients with more severe disease than indicated by the TG13 criteria was significantly higher than that of other patients [12]. In a multicenter study performed in Japan and Taiwan, all grade III prognostic factors (apart from hepatic dysfunction) that were significant in multivariate analysis had roughly equal weights [33]. Another study found that intra-hepatic biliary stenosis and hypoalbuminemia predicted poor prognosis, whereas renal failure, hepatic dysfunction, AC accompanied by malignant disease, and hypoalbuminemia did not [34]. We found that the TG18 grade independently predicted AKI, and that AC severity (as defined by TG18) was positively correlated with AKI severity. A TG grade of III also predicted 1-year mortality. Our study is unique in that we are the first to define the incidence, risk factors, and clinical outcomes of AKI associated with AC.

AKI is an increasingly common problem associated with adverse short- and long-term outcomes. Even modest elevations in serum creatinine levels were associated with adverse outcomes in this study; an increase of ≥ 0.5 mg/dL was associated with a 6.5-fold increase in the risk of death. AKI was also significantly associated with mortality, length of hospital stay, and healthcare costs in the short term [14]. The two major negative long-term outcomes of AKI are higher mortality [35, 36] and the development of CKD after discharge [37–39]. A recent systematic review on long-term prognosis after AKI demonstrated that the development of CKD after AKI (compared to AKI alone) doubled the mortality rate and was associated with a 4-to-5-fold increase in adverse CKD outcomes [40]. We found that the in-hospital mortality rate was higher in the AKI than non-AKI group. During a mean follow-up of 24.5 months, 1-year mortality was also higher in the AKI group. Cox's regression analyses showed that AKI was an independent predictor of all-cause mortality in AC patients after adjusting for all covariates. During a mean follow-up of 3.4 years, AKI development was associated with 1-year mortality (similar to previous studies) [36, 37, 40] and CKD progression [39, 40], according to the decline in eGFR.

We found that age, HT, TG grade III, and SIRS and qSOFA scores ≥ 2 predicted AKI in AC patients; the most influential factor was Tokyo Guidelines grade III. We developed a predictive AKI score based on these factors. This system sensitively and specifically predicts AKI in AC patients when using a cutoff ≥ 1.9. This allows clinicians to identify AC patients at high

risk of AKI in the emergency room or ICU, and to schedule specific treatment. However, further studies are needed.

Our study had both strengths and limitations. First, as this was an observational, retrospective single-center study, the results should not be generalized. Also, we cannot entirely rule out selection bias. Second, we could not identify other possible causes of AKI because we relied on medical records. Third, we used the serum creatinine cutoff of the KDIGO guidelines to diagnose AKI, because urine output was not measured in most patients. Therefore, AKI incidence and severity may have been underestimated. However, we believe that these limitations may have been overcome by the large number of patients. The laboratory data were well-documented and consistent, and management was unform (because this was a single-center study). This is the first large-scale study with a high number of patients to describe the incidence, risk factors, and clinical outcomes of AKI in patients with AC. A further well-designed prospective study is needed for validation.

## 5. Conclusions

In this large retrospective study, we showed that age, HT, TG grade III, and SIRS and qSOFA scores $\geq$ 2 significantly predicted AKI in AC patients, which was associated with both in-hospital and 1-year mortality and rapid renal function decline. Our predictive AKI scoring system may aid clinicians who manage AC patients.

## Supporting information

**S1 Fig. Calibration plot showing that predicted probability and actual probability are almost identical).**
(TIF)

**S1 Data.**
(XLSX)

**S1 Checklist.**
(DOC)

## Acknowledgments

The authors would like to thank Rock Bum Kim (Regional Cardiovascular Disease Center, Gyeongsang National University Hospital, Jinju, South Korea) for invaluable help in statistical analysis.

## Author Contributions

**Conceptualization:** Dong Jun Park.

**Data curation:** Tae Won Lee, Wooram Bae.

**Formal analysis:** Tae Won Lee, Wooram Bae, Dong Jun Park.

**Methodology:** Seongmin Kim, Jungyoon Choi.

**Writing – original draft:** Tae Won Lee.

**Writing – review & editing:** Eunjin Bae, Ha Nee Jang, Se-Ho Chang, Dong Jun Park.

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
