## [Decision Letter · Decision Letter 0]

10 Mar 2022

PONE-D-22-04137Incidence, risk factors, and prognosis of acute kidney injury in hospitalized patients with acute cholangitisPLOS ONE

Dear Dr. Park,

Thank you for submitting your manuscript to PLOS ONE. After careful consideration, we feel that it has merit but does not fully meet PLOS ONE’s publication criteria as it currently stands. Therefore, we invite you to submit a revised version of the manuscript that addresses the points raised during the review process.

We look forward to receiving your revised manuscript.

Kind regards,

Mabel Aoun, MD, MPH

Academic Editor

PLOS ONE

Journal Requirements:

“None”

“None”

6. Please include a separate caption for each figure in your manuscript.

Reviewers' comments:

Reviewer's Responses to Questions

**Comments to the Author**

1. Is the manuscript technically sound, and do the data support the conclusions?

Reviewer #1: Yes

Reviewer #2: Yes

Reviewer #3: Yes

2. Has the statistical analysis been performed appropriately and rigorously? 

Reviewer #1: Yes

Reviewer #2: Yes

Reviewer #3: Yes

3. Have the authors made all data underlying the findings in their manuscript fully available?

Reviewer #1: Yes

Reviewer #2: Yes

Reviewer #3: No

4. Is the manuscript presented in an intelligible fashion and written in standard English?

Reviewer #1: Yes

Reviewer #2: Yes

Reviewer #3: Yes

5. Review Comments to the Author

Reviewer #1: Dear Editor, Thank you for the privilege of reviewing this manuscript. This is a well designed and nicely written study that describes the statistics of AKI in acute cholangitis patients.

I suggest this manuscript to published in its present form.

Best regards,

Reviewer #2: This is an interesting paper for several points: the use of KDIGO criteria for AKI, number of patients and the long follow up after discharge. Authors are just kindly requested to move AKI definition to the introduction

Reviewer #3: In this study, Lee et al used a retrospective cohort design to investigate the incidence, risk factors, and prognosis of acute kidney injury in 1438 patients hospitalized with acute cholangitis (AC) in a tertiary care center () from January 2011 to December 2017. The investigators found that age, history of hypertension at baseline, AC severity criteria (Tokyo Grade score, SIRS and qSOFA) were independent predictors of AKI in AC patients. They also report that AKI was independently associated with all-cause mortality, and they developed a prediction model for AKI in AC patients, that had good discriminant properties (C-statistic = 0.798) and the sensitivity and specificity of the scoring system at a cut-off of 1.9 were 70.1% and 77.4% respectively.

The strengths of this study include access to high quality inpatient data in a large number of patients with AC to be able to answer the research question but there are several concerns that need to be addressed before this manuscript can be considered for publication.

Major concerns

1. Variable selection for AKI prediction model. The authors mention using forward and backward stepwise procedures to select the predictors included in the prediction model. Can the authors comment on the reproducibility of these variable selection procedures? Would a different set of variables not be selected if this study was repeated in another sample? Perhaps deciding a priori to select variables known to be predictive of AKI based on prior data and fitting the model once may be produce a more reproducible model? Also, would be more advantageous to model age in a nonlinear fashion using restricted cubic splines?

2. Validation of prediction model. Did the authors use any methods to validate their prediction model such as cross-validation or bootstrapping? In the absence of any validation, can the authors comment on how this model would perform in another dataset?

3. Model calibration. The authors present AUROC values for discriminant analysis but it would be important to know if model calibration was also assessed.

4. Lag time for AKI diagnosis and Time to death. Can the authors describe if there was any lag time between study entry (on date of admission) and the date of AKI diagnosis? If so, please present the median and interquartile range. Also, if any such lag time exists for some patients, how was the time pre-AKI diagnosis modelled in the Cox proportional hazards models for all-cause mortality. Was that time attributed to unexposed person-time?

Minor concerns

1. Figure 1. In the methods, patients were admitted between January 2011 and December 2017. The Figure says 2016. Please correct.

2. Figure 2. Please include the AUROC (95% CI) on the graph.

3. Figure 3. Please clarify what the time axis represents. Is it time from cohort entry to end of follow-up?

4. Table 2. It may be better to present the multivariable odds ratios for all the variables in the table.

5. Table 3. In order to make inferences about the frequency of severe AKI in the different Ac severity categories, it would be better to prevent the row percentages.

6. Discussion (page 10, paragraph 2). The percentages for AKI stages (56%, 28% and 16%) cannot be interpreted as AKI rates. Please revise. Also please ensure that all data presented in the discussion is also presented in the results.

7. Please clarify in the manuscript text what short-term and long-term mortality means.

6. PLOS authors have the option to publish the peer review history of their article (what does this mean?). If published, this will include your full peer review and any attached files.

Reviewer #1: **Yes: **Tuncay Sahutoglu, M.D.

Reviewer #2: No

Reviewer #3: No

---

## [Author Response · Author response to Decision Letter 0]

28 Mar 2022

We appreciate your positive comments. Files responding to reviewer and editor comments are attached. Please check it.

---

## [Editor Report · Decision Letter 1]

1 Apr 2022

Incidence, risk factors, and prognosis of acute kidney injury in hospitalized patients with acute cholangitis

PONE-D-22-04137R1

Dear Dr. Park,

We’re pleased to inform you that your manuscript has been judged scientifically suitable for publication and will be formally accepted for publication once it meets all outstanding technical requirements.

Kind regards,

Mabel Aoun, MD, MPH

Academic Editor

PLOS ONE
---

## [Editor Report · Acceptance letter]

5 Apr 2022

PONE-D-22-04137R1 

Incidence, risk factors, and prognosis of acute kidney injury in hospitalized patients with acute cholangitis 

Dear Dr. Park:

I'm pleased to inform you that your manuscript has been deemed suitable for publication in PLOS ONE. Congratulations! Your manuscript is now with our production department. 

Kind regards, 

on behalf of

Dr. Mabel Aoun 

Academic Editor

PLOS ONE